# The Role of Social Media in Food Product Choices Made by Polish Consumers

**DOI:** 10.3390/nu17111801

**Published:** 2025-05-26

**Authors:** Agnieszka Godlewska, Anna Katarzyna Mazurek-Kusiak, Andrzej Soroka

**Affiliations:** 1Faculty of Medical and Health Sciences, University of Siedlce, B. Prusa 14 St., 08-110 Siedlce, Poland; godlewskaa@uph.edu.pl (A.G.); andrzej.soroka@uph.edu.pl (A.S.); 2Department of Tourism and Recreation, University of Life Sciences in Lublin, Akademicka 15, 20-950 Lublin, Poland

**Keywords:** differences, woman, man, healthy lifestyle, purchase, nutrition, Internet

## Abstract

**Background:** The aim of the study was to determine the impact of social media on the food purchasing decisions and dietary attitudes of Polish consumers. **Methods:** The research was conducted using the diagnostic survey method, employing an online questionnaire technique CAWI (Computer Assisted Web Interview). A total of 1099 adults participated in the study, including 54.23% women and 45.77% men. The survey data were analysed using multidimensional discriminant analysis. **Results:** The study demonstrated that social media significantly affects the purchasing decisions of Polish consumers regarding food products. The study indicated that there is a gender difference in the impact of social media. Women more often use SM to gain knowledge about food products and shape their dietary attitudes, and they are more susceptible to marketing content. **Conclusions:** It seems necessary to regulate online advertising and eliminate excessive advertising, as this form of publication in the media is the most effective and has a significant impact on the purchase of food products by Polish consumers. Unrestricted access to content and lack of regulation lead to misinformation, which can result in erroneous purchasing choices.

## 1. Introduction

According to social learning theory [1], people learn behaviours by following others, especially social models such as influencers. On social media (hereinafter referred to as SM), users follow, comment on, and imitate the dietary choices presented by individuals who they trust or consider authorities. As a result, this medium has become a significant part of daily life, and its ubiquity affects the behaviours and functioning of the users [2,3]. SM is becoming an important and, in many cases, the sole source of information regarding the purchase of products and services, dictating and shaping societal preferences [4].

In 2023, more than 4.76 billion social media users were recorded globally, accounting for nearly 60% of the world’s population [5,6]. In Poland, nearly 75% of adults use SM. The dynamism of this phenomenon is evidenced by the fact that in 2023 in Poland, over 400,000 new users joined, and still counting [7].

Ajzen [8] demonstrated in the theory of planned behaviour that decisions regarding food purchases can be predicted based on intentions which are shaped by attitudes, social norms, and perceived behavioural control. SM influences all these elements by promoting specific norms and patterns, and impacting the perception of behaviours as socially desirable [8].

SM relies on interactive digital technologies which facilitate the creation and sharing of information, ideas, professional interests, or collaboration with the community [9] through online social networking outlets [10], which include Facebook, Instagram, YouTube, and TikTok [11]. By means of these tools, marketers can collaborate with consumers by leveraging user-generated content to increase brand awareness [12].

SM exerts a very strong, both planned and unplanned, educational influence on public health. It shapes desirable health behaviours and serves as a widespread source of accessible and attractive health-promoting knowledge. Through engaging and diverse forms of communication, SM makes the process of acquiring and broadening knowledge about a healthy lifestyle, as well as adopting positive behavioural patterns in this regard, enjoyable and more effective [13]. In this context, SM serves two functions:Educational—providing essential health-related information in a way that makes specialised knowledge understandable to the average citizen;Motivational—encouraging audiences to change their existing attitudes and behaviours to more health-promoting ones [14].

The model of knowledge transfer in food product choices combines the following three elements:Advertising ‘healthy’ food products, aimed at positively impacting the health of recipients and shaping their attitudes by promoting new, desirable values;Thematic activities, e.g., coverage of sports events or culinary competitions which subtly incorporate content about healthy eating and promote a healthy lifestyle;Showcasing the behaviours of ‘influencers’ who consume only healthy products [15].

Meeting Polish consumer needs, particularly food product-related ones, requires the creation of accurate, high-quality content on SM that showcases services and products. Information disseminated through SM sites can substantially affect Polish consumers’ purchasing decisions and dietary attitudes, making them innovative tools for obtaining and sharing information [16]. These topics are typically addressed by bloggers [17], but also by organisations focused on health promotion, and regular users of these sites.

On the other hand, much of the content provided by SM is actually hidden advertising. This type of superficial and general education is, in essence, merely a backdrop for selling certain products. The amount of false and misleading health-related information online is increasing year by year. According to data from the Institute of Media Monitoring in Poland, in June 2022 alone, social media contained approximately 3.6 thousand publications about dietary supplements, of which only 3% were statements by specialists [18]. The abundance of information, often contradictory and controversial, makes it difficult to assess its credibility and reliability [13], which can lead to confusion [19]. Distinguishing true and false information can be challenging, as most Polish SM users do not read the entire content of a post or check the provided information in other sources [20]. Popularity is not synonymous with competence. A significant portion of nutritional advice comes from individuals without relevant education, and the content is often based on simplifications, nutritional myths, or pseudoscience. Furthermore, SM algorithms deliver content aligned with users’ previous interests, which can reinforce erroneous beliefs, and hinder access to reliable information [21]. Generally, consumers spend little time thoroughly exploring information about specific food products [22]. According to Goffman’s Framing Theory [23], the way content is presented on SM, such as ‘superfoods’, ‘detox’, or ‘fit lifestyle’, can influence perceptions of health and food—e.g., perceiving certain products as ‘better’ or ‘trendy,’ regardless of their nutritional value, simply because they are promoted by famous individuals. Observing influencers who present an idealised ‘fit lifestyle’ can create social pressure and stress, which in turn negatively affects one’s relationship with food and body image [24,25].

Many consumers are motivated to interact with other users of social networking platforms [26], yet some distrust SM due to, e.g., an overload of unnecessary, low-value, or false information, including a large amount of spam [27].

People obtain nutritional information both through media and interpersonal sources. In their search for information, they participate in passive and routine SM use [28,29]. Access to nutritional information is now nearly universal via the Internet, with information varying in terms of scientific integrity and the expertise of the provider [30,31].

Researchers are increasingly interested in food-related issues, particularly those associated with nutrition and health, on SM [16,32,33,34,35]. However, a full understanding of the effect of SM on the food purchasing behaviours of consumers remains insufficient [36]. Purchasing decisions involve the evaluation of various factors including needs, preferences, product information, price, and quality [37]. Food choices are also influenced by individual consumer characteristics, such as gender, advertising [38], age and life stage, occupation, financial status, and lifestyle [39]. Purchasing decisions directly affect dietary habits and are an important topic in the context of health prevention and education, this being an area of interest not only to scientists but also to a broader audience.

The study reported here will add to the subject literature on the influence of social media on the food purchasing decisions of Polish consumers, and will determine their dietary attitudes.

## 2. Objectives and Research Methods

### 2.1. Research Objectives and Hypotheses

The aim of the study was to determine the impact of social media on the food purchasing decisions and dietary attitudes of Polish consumers. For the purposes of the study, after analysing the relevant literature [20,40,41,42] and conducting their own observations, the authors assumed the following hypotheses:

**H1:** 
*Exposure to nutritional content on SM positively correlates with the intention to purchase specific food products among Polish consumers, in line with the assumptions of the Theory of Planned Behaviour.*


**H2:** 
*SM users who actively seek food-related information are more sensitive to the price of food products than passive users.*


**H3:** 
*Women are more likely than men to use SM as a source of information about the nutritional value of products.*


**H4:** 
*Women, more frequently than men, shape their dietary attitudes under the influence of content posted on SM.*


**H5:** 
*Women are more likely than men to respond emotionally and behaviourally to the visual and narrative form of food-promoting content on SM, which influences their purchasing intentions.*


### 2.2. Materials and Methods

The study employed the diagnostic survey method, using an original questionnaire designed by the authors with 14 questions, of which five were used in this study:−To what extent do you use SM to obtain information about dining establishments and food products?−To what extent does a specific form of publication on SM encourage you to choose a particular food product?−For you, what are the most important factors in deciding to choose food products after viewing content about them on SM?−What information posted on social media most significantly influences the purchase of food products?−What is the impact of SM on your dietary attitudes?

To measure attitudes, a five-point Likert scale was applied (where 1 = low importance and 5 = high importance), preceded by a construction and validation procedure. The reliability of the scale was calculated, with Cronbach’s alpha value reaching 0.87. The Likert scale was used due to its ease of completion, which encourages respondents to participate in surveys, as well as its simplicity in analysis and interpretation.

Participants in the study were not time-restricted and were informed about the purpose of the study and assured of the anonymity of their data before completing the questionnaire. Contact details of the researcher were included in the questionnaire to allow respondents to obtain additional information if needed.

The study was conducted in Poland from May to July 2024. A total of 1099 adults participated in the study, including 596 women (54.23% of all the respondents) and 503 men (45.77% of all the respondents). Among the participants, 37.30% were aged up to 35 years, 50.59% were in the 36–60 age range, and 12.11% were aged 61 and older. This age division was chosen, firstly, due to ‘young adulthood’ being commonly defined as a unique life stage involving the transition from the dependency of adolescence, living with family and attending school, to independence and self-sufficiency [43]. The 61-and-older range marks the beginning of retirement age for women and the approaching retirement period for men, marking the age of withdrawal from productive activity. Regarding education, 79.45% of the respondents had a degree, 16.10% were secondary school-leavers, and 3.45% had vocational education.

Quota sampling was used to determine the size of the representative sample. The number of respondents was determined proportionally to the entire adult population of Polish women and men, taking into account the age and gender of the participants [44]. To calculate the sample size, a sample size calculator was used, which considered the total population of adult residents of Poland, accounting for the gender of the participants. This was also used to calculate the fraction, set at 0.5, with the confidence level of 95% and the maximum error of 5%. An application of this procedure indicated a minimum sample size of 1066. During the process of recruitment of respondents, quota sampling was used with the following criteria: at least 18 years of age, Polish nationality, and proportional selection taking into account the gender of the participants. The survey was conducted online using the Google Forms platform, which offers an intuitive interface allowing for quick questionnaire creation. The questionnaire was sent to respondents via the university email service (e-mail) and through links shared on social media platforms (Facebook, X, and Instagram), which significantly streamlined the data collection process. The survey was closed after receiving 1250 responses, and, after checking and exclusion of incomplete submissions, 1099 responses were retained for statistical analysis.

Statistica 13.1 PL (StatSoft Inc., Tulsa, OK, USA) was used for statistical analyses and the implemented discriminatory function, which was used to resolve which variables discriminated two emerging groups. Also, the purpose of applying this method was to identify differences between factors significant for group membership. The classification function was applied in the form of coefficients that were defined for each group. Prior to the analysis, multivariate normality was analysed, testing each variable for normal distribution. It was assumed that variable variance matrices are homogeneous in groups. Tabachnick and Fidell [45] suggest that with a large number of respondents, as in this study, minor deviations are of little importance. The use of Wilks’ Lambda procedure in the discriminatory function analysis aimed to select variables which, when introduced into the equation, would minimise its value to the greatest extent. Means for which the probability was less than *p* < 0.05 were considered statistically significantly different [46].

## 3. Results

The analysis of the most popular SM used to provide Polish consumers with knowledge about food products highlights Instagram as the site most frequently utilised by respondents. Women use this platform to a greater extent, as demonstrated by the developed discriminatory function model, where the classification function for women was significantly higher (at *p* = 0.029) than for men. Similarly, the second most popular SM site, Facebook, was also more popular with women. A statistically significant difference (at *p* < 0.001) was observed, with women showing a higher mean value compared to men. Other SM platforms, such as YouTube, and the X platform had considerably less significance (Table 1).

Advertisements for specific food items were the most effective publicity form on SM that significantly influenced the purchase of food products, as pointed out by respondents. As revealed by value of the classification function, the created discriminatory function model identified this form as being more frequently reported by women than men. For women, the classification function value was significantly higher (at *p* < 0.001) than for men. For men, published photos of a given product had a greater influence on purchasing decisions. The average classification function value in this case was 1.266, significantly higher (at *p* < 0.001) than for women. An important factor influencing the purchase of a specific food product was the number of likes under a product’s post. Females responded to this form significantly more often with a classification, compared to for males, showing a significant difference at *p* < 0.001. The model also included, albeit with considerably lower classification function values, other forms such as stories about a given food product and videos related to the product, to which women responded significantly more readily (at *p* < 0.001), as well as text messages about products, which men used significantly more often (at *p* < 0.001) (Table 2).

Women’s greater susceptibility to advertisements, the number of likes, or influencer recommendations may be related to their stronger social orientation and tendency to engage with communal and emotional content [11,47]. In contrast, men are more often driven by visual elements (photos) and specific textual information, which may stem from a more task-oriented decision-making style in purchasing [48]. Furthermore, the variables with the highest loadings in the discriminatory functions (e.g., ‘viewed advertisements’, ‘number of likes’, ‘weight-loss properties’, ‘product quality’) suggest that SM interventions operate through persuasive mechanisms, heuristics of information processing (e.g., the ‘social proof effect’), and the influence of social norms, which is supported by the literature [10,49,50].

Price was the most important factor of those considered when choosing a specific food product. Women paid more attention to this aspect (at *p* < 0.001) than for men. Men wanted to know more about the product’s origin, as indicated by a classification function value, which was significantly higher (at *p* < 0.001) than for women. At a slightly lower but similar level in both groups, the importance was raised of factors related to the health aspects of purchasing a given product and the location of purchase. The model also included factors concerning the product’s packaging and aspects of its environmentally friendly production, which were significantly more important to males (at *p* = 0.016 and *p* < 0.001, respectively). By contrast, females placed significantly greater emphasis (at *p* < 0.001) on the originality of the offered products (Table 3).

The information available on SM regarding product quality was found to be the most important, and this applied to both test groups. However, it was men who indicated that this information is significantly more important for them than for women. The classification function value for men was significantly higher than for women.

Also, males pointed to the information that the food product was produced in the vicinity as significantly more important, with *p* = 0.005, than females. The created model also included information regarding the slimming properties of the product. With *p* < 0.001, this information was significantly more important for women than for men (Table 4).

The nutritional attitude influenced by SM varies depending on gender. The study showed that men are significantly more likely to purchase a given product under the influence of SM compared to women. In their case, the classification function value which, with *p* < 0.001, was significantly higher than the value declared by women. On the other hand, women are significantly more likely to change their eating habits under the influence of SM than men. Women also try to use diets promoted and recommended on SM to a significantly higher degree, with *p* = 0.021, respectively (Table 5).

## 4. Discussion

The results of the study reported here provide significant insights into the impact of SM on the purchasing decisions of Polish consumers buying food products, and they align with a broader trend of research on consumer behaviour in the digital era [51,52].

In light of the proposed hypotheses, several key relationships and gender-related differences can be observed regarding the use of SM to obtain information about food products and make related purchasing decisions.

The study indicated that, in addition to the influence of peer groups, family, and traditional media, Polish consumers are increasingly exposed to information via SM, which impacts social norms and behaviours [53] such as food product purchasing decisions. It aligns with a common global trend of using SM as a tool for engaging consumers [54]. This is confirmed by a study of the American population, which showed that over 72% of Americans use SM to communicate and share texts, images, and videos [55]. Presence on SM is important for users with various motivations, such as increasing brand value, selling goods and services, or deriving financial benefits from a large number of followers [56]. Although the multidimensional discriminant analysis used in the study was statistical in nature and does not allow for direct modelling of cause-and-effect relationships, these results enable the formulation of hypothetical mechanisms through which social media influence consumers’ purchasing decisions and dietary attitudes. It can be assumed that the impact of SM is, e.g., through psychological mechanisms related to the processing of social cues such as the number of likes (popularity heuristic), the presence of influencers (modelling and social proof mechanisms), or the visual appeal of content (sensory and emotional perception) [10,47,50]. Women, who exhibit greater susceptibility to emotional and social messaging, more frequently respond to marketing content, whereas men are more responsive to functional and visual aspects, which is reflected in the variables that best differentiate groups in the discriminant analysis [11,48]. SM users generate billions of views of food-related content, and many feel an urge to consume the exact type of food they have seen on SM. It should also be noted that socially endorsed images of food can increase the consumption of those products [48], which aligns with our findings. According to the obtained results, it was also found that published photos of food products significantly affect consumers’ purchasing decisions, particularly among men. The present study confirmed that women use SM more frequently to gain knowledge about food products, and they significantly more often use the two most popular platforms in Poland, Instagram and Facebook. Additionally, the research revealed that Instagram is the most frequently used site for such purposes, particularly among women. Facebook also plays an important role in the purchasing process, as demonstrated by significant statistical differences in its use by women and men. Other options, such as YouTube, TikTok, and X, play a smaller but still noticeable role. These results are consistent with earlier studies by Dwivedi [57] and Voorveld [58], who demonstrated that SM platforms have become an important element in consumers’ daily lives. According to these authors, Facebook is the most popular of all the SM platforms.

The study also confirmed that SM users pay much attention to the price of the food products they seek and purchase. Price proved to be a key factor influencing the choice of food products, it being more significant for women than for men. Additionally, men paid more attention to the product’s origin and ecological aspects, while women were more interested in the originality of products. These findings are consistent with earlier studies by other authors [59,60] indicating that price and quality are key determinants of consumer behaviour. The results of the study by Antoniak et al. [61] also confirm that attributes such as reasonable and low prices positively influence consumers’ perception of products. Meanwhile, studies by Cao and Miao [62] showed that consumers attach great importance to the health and ecological aspects of purchased products. This is further supported by studies conducted by a team of researchers from the USA, who pointed to a preference among American consumers for organic, minimally processed food produced using minimally processed raw materials without unnecessary additives—an approach particularly displayed by elderly individuals, women, and degree holders [63]. Purchasing decisions are influenced by consumers’ perceptions of the environmental impact of the production of purchased goods and the use of ecological methods in the production process [64]

Females are more responsive to various publication forms such as advertisements and number of likes under posts about food products, which may indicate their greater susceptibility to marketing messages. By contrast, males more often make purchasing decisions based on published photos and text messages about products. These findings correspond with earlier reports by Vandenbosch and Eggermont [65] on the influence of gender on online consumer behaviour. Furthermore, there is a gender-related difference in the impact of social media. Women more often use SM to find out about food products and shape their nutritional attitudes. Research emphasises the importance of consumer education to reduce uncertainty during purchases, and encourage more informed choices, making education a necessary and essential element, including avoiding GMOs. Improving consumer education is necessary to reduce uncertainty and promote more conscious choices [66]. Sá et al. [67] identified three main factors influencing purchasing decisions: natural nutrition, product quality, and nutritional value. The analysis of the study results confirmed the importance of product packaging and labels, which typically display, among other things, protein source or carbohydrate content, and contribute to consumer awareness of environmentally friendly products, thereby creating demand for organic and healthy food. Similar research results were obtained by Plasek et al. [68], whereas Vidal et al. argue that using information provided by product labels, particularly those emphasising low calorie content and beneficial nutrients, is an effective marketing strategy for promoting healthy eating behaviours [69]. Healthy food consumption habits refer to practices and behaviours related to choosing and consuming food that is minimally processed and free from artificial additives, preservatives, and contaminants. These habits emphasise the consumption of whole, natural products such as fruits, vegetables, whole grains, lean proteins, and healthy fats. They focus on food safety, nutrition, and the overall quality of ingredients, often including preferences for organic and locally sourced products [70]. There are many factors in product attributes that influence healthy eating behaviours, including personal factors such as health consciousness and understanding product label information [71,72,73]. Moreover, healthy consumption habits involve reducing the intake of unhealthy ingredients such as saturated fats, added sugars, and sodium combined with increased consumption of fibre, vitamins, and minerals. This holistic approach to nutrition is often based on dietary guidelines and recommendations from health organizations [74,75].

Women demonstrated a significantly higher demand for various diets proposed by SM and were more willing than men to change their eating habits under the influence of SM. However, men were more likely to purchase a product promoted on SM. This has also been confirmed by studies by Leksy and Nowak [76] and Zeeni et al. [77], which revealed that SM significantly influence consumers’ nutritional attitudes: men more often make purchases under the influence of content published on social media while women are more likely to change their eating habits and try diets promoted on these platforms. This may suggest that women treat SM as a source of information and inspiration for a healthy lifestyle, while men make more impulsive purchasing decisions.

Similar results were obtained in studies on the impact of SM on health and dietary behaviours conducted by Chung et al. [47], who noted that SM users, especially females, are more likely to engage in health strategies promoted by influencers and other public figures. Their research suggests that SM effectiveness in promoting a healthy lifestyle may partly result from the social identity and norms created by these sites. Also, Alfeel and Ansarii [78] point to the fact that the perception of the authenticity of content published on SM is a key factor influencing customer purchasing decisions. Their results indicate that men may approach promoted diets and products on SM more sceptically, which could explain their lower interest in changing eating habits. By contrast, women, who are more inclined to seek support in their pursuit of a healthier lifestyle, may be more open to such influences. Furthermore, studies by Możdżonek and Antosik [79] revealed that SM have a significant impact on positive changes in eating habits, particularly among women. Meanwhile, men, due to a short-term approach to shopping and a tendency to choose products in attractive packaging, may not fully appreciate the long-term benefits of more complex and healthier dietary choices.

## 5. Summary and Conclusions

Social media are becoming an increasingly popular research topic, yet there is still poor understanding of their role in the choice of food products and nutritional attitudes. Our study fills a gap in the existing literature on behavioural patterns related to the purchase of food products and the nutritional attitudes of Polish consumers as influenced by social media.

The obtained results confirmed that SM significantly influence the purchasing decisions of food products among Polish consumers. The study revealed gender differences in the impact of social media. Women utilise SM more often to gain knowledge about food products and shape their dietary attitudes, and they are more susceptible to marketing content. The study demonstrated that consumers engaged with SM are particularly influenced by posted advertisements and photos of food products, as well as the number of likes, which undoubtedly affects their purchasing decisions. Unrestricted access to content and lack of regulation lead to widespread misinformation, which can result in erroneous purchasing choices. This highlights the need for consumer education on healthy eating to enable them to make informed purchasing decisions and avoid the influence of unreliable information on SM.

In light of the above findings, food-related regulations should be strengthened. It is necessary to increase the accuracy of product information by regulating online advertising and eliminating excessive advertising, as this form of publication on SM is the most effective and has an enormous impact on the purchase of food products by Polish consumers. The price of the product was found to be the most significant factor considered when choosing a food product, fully confirming the hypothesised assumption.

These results can serve as valuable guidance for marketers and companies involved in the sale of food products, who should tailor their marketing strategies to the gender-specific characteristics of their customers. Further research could focus on analysing other psychological and social factors that may additionally influence purchasing decisions in the digital environment.

## 6. Limitations and Future Research Directions

The study has several limitations that present opportunities for future research. Firstly, in addition to consumer interest, investors, national policies, and the broader online SM audience may also be relevant. Secondly, this topic has been the subject of few studies so far, making it another direction for future analysis. Thirdly, the obtained results refer to food products on SM and should not be generalized to other products or industries, which implies another direction for research. Moreover, in the future, research may be conducted using innovative methods (e.g., experimental design, longitudinal studies) or cross-cultural comparisons.

## Figures and Tables

**Table 1 nutrients-17-01801-t001:** The importance of social media in gaining information about food products.

Social Media	Wilks’ Lambda: 0.457 F = 18.048 *p* < 0.001			Classification Function	
	Wilks’ Lambda	F Value	P Level	Women	Men
Instagram	0.458	4.756	0.029 *	2.659	2.411
Facebook	0.498	8.352	0.001 *	1.270	1.130
YouTube	0.512	1.167	0.283	0.678	0.705
Tik Tok	0.478	34.138	0.001 *	0.621	0.256
Instagram X (former Twitter)	0.489	4.104	0.043 *	0.387	0.279
constant				4.786	6.018

*—significant difference at *p* < 0.050.

**Table 2 nutrients-17-01801-t002:** Publicity forms on social media affecting the choice of food products.

Publicity Forms on SM	Wilks’ Lambda: 0.539 F = 27.564 *p* < 0.001			Classification Function	
	Wilks’ Lambda	F Value	P Level	Women	Men
Advertisements encouraging product purchase	0.543	15.389	0.001 *	1.721	1.284
Published photo of the product	0.578	20.645	0.001 *	0.976	1.266
number of likes under a post about the product	0.532	37.586	0.001 *	1.021	0.626
Stories about the product	0.561	10.828	0.001 *	0.694	0.475
Text messages about the product	0.521	33.834	0.001 *	0.110	0.501
Videos about the product	0.563	50.487	0.001 *	0.304	0.136
constant				5.056	5.768

*—significant difference at *p* < 0.050.

**Table 3 nutrients-17-01801-t003:** Factors affecting food product purchase.

Factors Affecting the Choice of Food Products	Wilks’ Lambda: 0.431 F = 24.440; *p* < 0.001			Classification Function	
	Wilks’ Lambda	F Value	P Level	Women	Men
Product price	0.412	13.010	0.001 *	2.151	1.887
Product’s origins	0.419	14.879	0.001 *	0.709	1.011
Health-related aspects of the product	0.387	1.232	0.265	0.842	0.921
Location of the purchase place	0.388	1.667	0.198	0.421	0.541
Product packaging	0.423	5.794	0.016 *	0.196	0.415
Originality of the offered products	0.441	30.777	0.001 *	0.652	0.237
Environmentally-friendly aspects of the product	0.392	52.331	0.001 *	0.200	0.366
constant				5.466	10.436

*—significant difference at *p* < 0.050.

**Table 4 nutrients-17-01801-t004:** Information published on social media affecting decision to purchase the foodstuff.

Kind of Information	Wilks’ Lambda: 0.584 F = 32.467; *p* < 0.001			Classification Function	
	Wilks’ Lambda	F Value	P Level	Women	Men
Quality of the given product	0.612	4.576	0.032 *	1.162	1.307
Product’s regionality	0.588	7.892	0.005 *	1.014	1.181
Slimming properties of the product	0.601	42.756	0.001 *	0.774	0.375
constant				5.443	6.476

*—significant difference at *p* < 0.050.

**Table 5 nutrients-17-01801-t005:** The impact of social media on respondents’ dietary attitude.

The Type of Influence of Social Media on Dietary Attitudes	Wilks’ Lambda: 0.548 F = 29.327; *p* < 0.001			Classification Function	
	Wilks’ Lambda	F Value	P Level	Women	Men
I buy products advertised on social media	0.528	65.342	0.001 *	0.376	0.915
I try out diets advertised and recommended by social media	0.578	5.265	0.021 *	0.856	0.699
I change my dietary habits under the influence of social media	0.542	53.736	0.001 *	0.883	0.374
constant				2.750	2.686

*—significant difference at *p* < 0.050.

## Data Availability

The original contributions presented in this study are included in the article. Further inquiries can be directed to the corresponding author.

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
