# Peer review of "The Role of Social Media in Food Product Choices Made by Polish Consumers"

_nutrients, 2025, doi:10.3390/nu17111801_

Round 1

Reviewer 1 Report

Comments and Suggestions for Authors

Overall evaluation
The manuscript is timely and relevant, addressing an important area of consumer research in the context of digital media. However, it needs a stronger theoretical framing, a clearer methodological justification, and a more critical interpretation of results. The gender focus is interesting but overplayed at the expense of other variables like platform type or information credibility. The manuscript would benefit from language polishing for clarity and coherence, as well as a tightening of the structure across sections to reduce redundancy.

Abstract
The abstract is generally clear and structured; however, it contains redundancy (e.g., “Abstract: Abstract:”) and could benefit from a more concise summary. The use of “CAWI” should be spelled out for international readers unfamiliar with this acronym. Also, the mention of specific statistical software (Statistica) is unnecessary in an abstract unless it provides insight into the novelty of the method. The gender comparison is overemphasized without sufficient theoretical framing, and the term “posted photos and textual information” lacks clarity. The abstract should end with a broader implication of the findings beyond gender differences.

Introduction
The introduction provides context and identifies the research problem clearly, but the narrative is somewhat unfocused and over-reliant on statistics and general claims about SM. It lacks a strong theoretical foundation or conceptual framework that could anchor the research. Citations are frequent but not critically integrated, and the motivation for focusing on Poland specifically is weakly articulated. The hypotheses are reasonable but should be reformulated more precisely and with reference to existing models of consumer behaviour or media influence. The third hypothesis, in particular, is too broad and combines multiple elements (knowledge seeking, attitudes, responsiveness to various forms of content) that should be broken down.

Materials and Methods
The methods section is generally well described in terms of survey design and statistical approach. However, the justification for choosing specific demographic segments is too vague. The description of quota sampling contradicts the actual process, which appears more convenience-based through SM platforms. The claim that “slight deviations were not that significant due to the large number of respondents” is methodologically weak and should reference power analysis or robustness tests. Also, more detail should be given on how the Likert scale items were developed and validated. The use of multivariate discriminant analysis is appropriate, but the rationale for its selection over other methods is missing.

Results
The results are detailed but read more like descriptive commentary than structured reporting. Key issues include over-reliance on classification function values without sufficient interpretation, as well as missing effect sizes and confidence intervals. The tables are helpful but poorly integrated into the narrative, most are simply referenced with minimal contextual explanation. The focus on gender differences, while consistent, becomes repetitive and lacks deeper analytical insight. More attention should be given to patterns across platforms, content types, or food categories. The interpretation of statistical significance is often confused with substantive importance.

Discussion
The discussion begins with a strong overview of the findings and connects them to existing literature; however, the argument becomes meandering. There is an overemphasis on confirming the hypotheses rather than critically engaging with unexpected findings or limitations. The discussion repeatedly conflates correlation with causation. Furthermore, while the gender-related insights are well developed, they are not interpreted through relevant gender or media theories. The paragraph structure is loose, leading to redundant and sometimes speculative claims. References are plentiful but not always well synthesised or critically evaluated.

Summary and conclusions
The conclusions effectively summarise the main findings but could be more concise. They lack a forward-looking orientation—there is little discussion of how marketers, policymakers, or educators can apply the insights. Points 5–7 read more like additional discussion than conclusions and should be integrated into earlier sections or reframed. The distinction between attitude change and actual behavioural outcomes is not well articulated.

Limitations and future research
The limitations section is brief and superficial. It fails to reflect on methodological biases such as self-selection, social desirability bias in self-reports, or lack of cross-validation of results. The suggestions for future research are too generic and miss an opportunity to propose innovative methods (e.g., experimental design, longitudinal studies) or cross-cultural comparisons.

Comments on the Quality of English Language

A thorough, line-by-line copy edit by a professional academic language editor is strongly recommended.

Author Response

REVIEW RESPONSE 1

Dear Sir/Madam

The authors would like to thank the Reviewer for valuable comments and suggestions contained in the review which will undoubtedly contribute to the quality of our article. We have responded to the reviewer's comments. We attach a revised version of the manuscript and responses to the Reviewer's comments.

We hope our answers and additions will be satisfactory responses to your questions and comments.

Thank you for considering this manuscript.

Abstract
The abstract is generally clear and structured; however, it contains redundancy (e.g., “Abstract: Abstract:”) and could benefit from a more concise summary. The use of “CAWI” should be spelled out for international readers unfamiliar with this acronym. Also, the mention of specific statistical software (Statistica) is unnecessary in an abstract unless it provides insight into the novelty of the method. The gender comparison is overemphasized without sufficient theoretical framing, and the term “posted photos and textual information” lacks clarity. The abstract should end with a broader implication of the findings beyond gender differences.

Response: The abstract has been revised as suggested by the Reviewer: the repetition (abstract, abstract) was removed, the meaning of the acronym CAWI was explained, the mention of specific statistical software (Statistica) was removed, it was briefly supplemented with a more extended summary).

Introduction
The introduction provides context and identifies the research problem clearly, but the narrative is somewhat unfocused and over-reliant on statistics and general claims about SM. It lacks a strong theoretical foundation or conceptual framework that could anchor the research. Citations are frequent but not critically integrated, and the motivation for focusing on Poland specifically is weakly articulated. The hypotheses are reasonable but should be reformulated more precisely and with reference to existing models of consumer behaviour or media influence. The third hypothesis, in particular, is too broad and combines multiple elements (knowledge seeking, attitudes, responsiveness to various forms of content) that should be broken down.

Response: Theoretical frameworks were added, including Social Learning Theory [Bandura 1977], Theory of Planned Behavior [Ajzen 1991], and Framing Theory [Goddman 1974]. Introduction was enriched with critical citations and greater reference to the Polish consumer. The hypotheses were improved, and the third hypothesis was divided into 3 separate ones.

Materials and Methods

The methods section is generally well described in terms of survey design and statistical approach. However, the justification for choosing specific demographic segments is too vague. The description of quota sampling contradicts the actual process, which appears more convenience-based through SM platforms. The claim that “slight deviations were not that significant due to the large number of respondents” is methodologically weak and should reference power analysis or robustness tests. Also, more detail should be given on how the Likert scale items were developed and validated. The use of multivariate discriminant analysis is appropriate, but the rationale for its selection over other methods is missing.

Response: The research sample elements were selected proportionally to the population of adult residents of Poland, taking into account the respondents' gender and age. The work was supplemented with a procedure for calculating the research sample using a sample size calculator which considered the total population of adult residents of Poland, including the gender of the respondents. This was also used to calculate the fraction set at 0.5, with the confidence level of 95%, and the maximum error of 5%. The minimum sample size was calculated to be 1066. The SM platform was used due to its widespread application, facilitating the collection of research material. The Likert scale was used due to its convenience for respondents who are more willing to participate in such surveys. The five-point Likert scale allows for a more precise representation of the attitudes studied, which is particularly important when it is crucial, as in this study, to detect even small differences that may affect the results. The authors relied on the research of Tabachnick and Fidell (1996) who state that with a large number of respondents, the deviation decreases and may have less significance. The purpose of applying multidimensional discriminant analysis was to identify variables that best differentiate groups, and to build a model that would accurately predict the membership of new observations in a given group.

Results
The results are detailed but read more like descriptive commentary than structured reporting. Key issues include over-reliance on classification function values without sufficient interpretation, as well as missing effect sizes and confidence intervals. The tables are helpful but poorly integrated into the narrative, most are simply referenced with minimal contextual explanation. The focus on gender differences, while consistent, becomes repetitive and lacks deeper analytical insight. More attention should be given to patterns across platforms, content types, or food categories. The interpretation of statistical significance is often confused with substantive importance.

Response: The description of the results has been revised by the authors.

Discussion
The discussion begins with a strong overview of the findings and connects them to existing literature; however, the argument becomes meandering. There is an overemphasis on confirming the hypotheses rather than critically engaging with unexpected findings or limitations. The discussion repeatedly conflates correlation with causation. Furthermore, while the gender-related insights are well developed, they are not interpreted through relevant gender or media theories. The paragraph structure is loose, leading to redundant and sometimes speculative claims. References are plentiful but not always well synthesised or critically evaluated.

Response: This section has been revised by the authors.

Summary and conclusions

The conclusions effectively summarise the main findings but could be more concise. They lack a forward-looking orientation—there is little discussion of how marketers, policymakers, or educators can apply the insights. Points 5–7 read more like additional discussion than conclusions and should be integrated into earlier sections or reframed. The distinction between attitude change and actual behavioural outcomes is not well articulated.

Response: Conclusions have been changed as suggested by the Reviewer.

Limitations and future research

The limitations section is brief and superficial. It fails to reflect on methodological biases such as self-selection, social desirability bias in self-reports, or lack of cross-validation of results. The suggestions for future research are too generic and miss an opportunity to propose innovative methods (e.g., experimental design, longitudinal studies) or cross-cultural comparisons.

Response: Limitations and future research have been expanded as suggested by the Reviewer.

Reviewer 2 Report

Comments and Suggestions for Authors

Please refer to my attached version that includes comments in the manuscript itself. 

Overall, the topic is timely, relevant, and interesting. The Methods, Discussion, and Conclusions require work.

Methods

The methods lack important details about the specific questions used including those that measured gender, age range, etc. They should be added. Other details about the statistical methods used and the purposes - for example a lambda procedure was used without context and only first appears in the tables.  There are other details lacking in the methods. The authors should be reminded that the methods should contain enough detail the someone reading them could replicate the study.

Results

The tables in the results don't have sufficient detail in the subheadings. Authors, recall that the reader should be able to interpret the table just by looking at it.

Discussion and Conclusions

The discussion and conclusion are basically a repetition of the results. I've included some comments in the paper.

In all, this is an interesting topic that could be worthy of publication if the recommended strengths are added.

Author Response

REVIEW RESPONSE 2

Dear Sir/Madam

The authors would like to thank the Reviewer for valuable comments and suggestions contained in the review which will undoubtedly contribute to the quality of our article. We responded to the reviewer's comments. We attach a revised version of the manuscript and responses to the Reviewer's comments.

We hope our answers and additions will be satisfactory responses to your questions and comments.

Thank you for considering this manuscript.

Methods

The methods lack important details about the specific questions used including those that measured gender, age range, etc. They should be added. Other details about the statistical methods used and the purposes - for example a lambda procedure was used without context and only first appears in the tables.  There are other details lacking in the methods. The authors should be reminded that the methods should contain enough detail the someone reading them could replicate the study.

Responses to the comments included in the work:

What is the name or source of this original questionnaire?

Response: It is the authors' original questionnaire (correction has been made in the text)

Somewhere, state the questions or the specific behaviors, etc. in the questions.

Response: This has been added in the chapter Materials and Methods

Clarify that the contact details of the researchers were provided.

Response: It has been added in the text according to the Reviewer's suggestion.

Were these traits part of the recruiting criteria? Presumably, this is a convenience sample. Please note that and include that this is a limitation.  Is there equal representations among geographic regions? 

Response: The assumption was to select a research sample that would proportionally represent the participation of the entire population of adult residents of Poland, taking into account the respondents' gender and age. To calculate the sample size, a sample size calculator was used which considered the total population of adult residents of Poland, including the gender of the respondents. This was also used to calculate the fraction set at 0.5, with the confidence level of 95%, and the maximum error of 5%. The minimum sample size was calculated to be 1066.

Does your institution require approval by an institutional review board?  If so, state that, as well as an approval number.

Response: Our Institution require no approval by an institutional review board.

Results

what units? Does mean that the mean for the women was 2.659 points greater than the mean for the men?  Also, report to 2 decimal places throughout.

Response: These values are averages, for example, the average for women was 2.659, and the average for men was 2.411. However, to avoid repetition in the discussion of results, values already included in the tables were removed.

Add a small note in the methods about what this measures.

Response: This was added in the chapter Materials and Methods.

Add more information so that readers can interpret the tables without looking at the text. Look at examples from other publications. No need to state the source of the data. 

Response: The tables seem to be quite clear as Wilks' lambda indicates the discriminatory power in the discriminant function, and F and P values explain the reasons for differences between groups. The tables contain the most important data which provide appropriate information about the analysis, presenting it very clearly.

It is implied in the text that these are the authors' data and analyses.

Response: Data sources were removed according to the Reviewer's suggestion.

Again, please include the questions. What items were they asked about - all items in every category or separate food categories?

Response: This was added in the chapter Materials and Methods

Discussion and Conclusions

These should be integrated into the discussion to support or refute your findings. As they are now, they fit better in the Introduction. The Discussion section is for discussing the reasons why your data makes or does not make sense and what it means, practically. Refer to other papers to support.

Response: This fragment was removed from the manuscript as suggested by the Reviewer.

which ones?

Response: It have been corrected according to the Reviewer's suggestion.

Same comment as above. This statement doesn't seem to directly connect to a specific finding.

Response: This statement was linked to the obtained research results.

Describe how your results are supported by theirs. "These results are consistent with ... who found that xyz as well."

Response: It has been corrected as suggested by the Reviewer.

Here is where you add support from other researchers in a very specific way. As is, it is more of a reporting of the results again.

Response: This fragment of Discussion was changed as suggested by the Reviewer.

Again, this section repeats what we've read above. What can you conclude about these findings? For example, should platforms change their approaches to attract more men than women?

Response: The conclusions were rewritten by the authors.

Reviewer 3 Report

Comments and Suggestions for Authors

The Role of Social Media in Food Product Choices Made by Polish Consumers

Although this study is quite interesting, it has major flaws:

Therefore, I recommend major changes before further processing:

1- A clear mechanistic explanation is missing that explains how intervention impacts the outcome.

2- Better to discuss the potential mechanisms involved, like biochemical and physiological.

3- well-designed control group could provide better support for the conclusions because the current control group is insufficient for the tested intervention.

 4- Statistical analyses are unclear. Please explain the description of the data and methods.

5- Sample size is limited, which also limits the main outcome.

6- Practical relevance in the results section is also limited. It's better to contextualise the results.

7- Include more literature and recent studies,, which will help you to highlight this study's contributions.

8- Better to improve the abstract.

9- Improve summary and conclusion.

10- Be consistent in using a same pattern of language and also improve grammar.

Comments on the Quality of English Language

Major improvement needed

Author Response

REVIEW RESPONSE 3

Dear Sir/Madam

The authors would like to thank the Reviewer for valuable comments and suggestions contained in the review which will undoubtedly contribute to the quality of our article. We responded to the reviewer's comments. We attach a revised version of the manuscript and responses to the Reviewer's comments.

We hope our answers and additions will be satisfactory responses to your questions and comments.

Thank you for considering this manuscript.

1- A clear mechanistic explanation is missing that explains how intervention impacts the outcome.

Response: We would like to clarify that our analysis was exploratory-descriptive in nature and relied on the use of multidimensional discriminant analysis which aimed to statistically identify variables differentiating purchasing decisions and dietary attitudes depending on gender in the context of social media (SM) use. This method does not model causality in a mechanistic sense but allows for the identification of variables which best distinguish the observed groups.

As you suggested, we have supplemented the manuscript with an interpretation of possible psychological-behavioral mechanisms that might account for the observed differences.

Although discriminant analysis does not make it possible to fully reflect the causal mechanisms, its results enable the formulation of mechanistic hypotheses for further research in which causal models will be applied.

Taking into account the reviewer’s comment, in the latest version of the manuscript, we have expanded the Discussion section to include an interpretation of potential mechanisms underlying the influence of content published on SM on consumer decisions and dietary attitudes.

Thus, although the study was not experimental in nature, its results allow for drawing conclusions about possible mechanisms of SM’s influence on consumers which can serve as a starting point for further, in-depth causal research. 

2- Better to discuss the potential mechanisms involved, like biochemical and physiological.

Response: We are grateful for this suggestion which will be of use in future research

3- well-designed control group could provide better support for the conclusions because the current control group is insufficient for the tested intervention.

Response: The study included the procedure for calculating the research sample. The sample size was determined as a proportional representation of adult residents of Poland, taking into account their age and gender. To calculate the sample size, a sample selection calculator was used, which considered the total population of adult residents of Poland, including the gender of the respondents. This was also used to calculate the fraction set at 0.5, with the confidence level of 95%, and the maximum error of 5%. The minimum sample size was calculated to be 1066. 

 4- Statistical analyses are unclear. Please explain the description of the data and methods.

Response: An extended explanation regarding the selection of statistical analyses has been included. The purpose of using the Lambda procedure was clarified, which aimed to select variables that, when introduced into the equation, most significantly reduce the coefficient, minimising its value.

5- Sample size is limited, which also limits the main outcome.

Response: The sample size was determined using a sample size calculator, which indicated the need to survey a minimum number of respondents, calculated at 1066. In the study, 1099 valid surveys were analysed.

6- Practical relevance in the results section is also limited. It's better to contextualise the results.

Response: The description of the results was revised by the authors.

7- Include more literature and recent studies,, which will help you to highlight this study's contributions.

Response: Additional recent literature has been cited.

8- Better to improve the abstract.

Response: Abstract has been changed as suggested by the Reviewer.

9- Improve summary and conclusion.

Response: Summary and conclusion have been improved as suggested by the Reviewer.

10- Be consistent in using a same pattern of language and also improve grammar.

We did our best to introduce the improvements while revising the whole manuscript.

Round 2

Reviewer 1 Report

Comments and Suggestions for Authors

The authors have addressed the majority of my comments in a satisfactory manner, and the revised manuscript shows substantial improvement in clarity, structure, and theoretical grounding. However, one important structural issue remains: the hypotheses are embedded within the Introduction. This is not in line with standard academic conventions. Hypotheses should be moved to a distinct section following the Introduction, typically titled “Research Objectives and Hypotheses” or similar. Including them within the Introduction disrupts the narrative flow and causes confusion in later references to the hypotheses. I recommend consolidating the hypotheses into a single, clearly numbered set and presenting them in a dedicated section outside the Introduction.

Author Response

REVIEW RESPONSE 1

Dear Sir/Madam

The authors would like to thank the Reviewer for valuable comments and suggestions contained in the review which will undoubtedly contribute to the quality of our article. We have responded to the reviewer's comments. We attach a revised version of the manuscript and responses to the Reviewer's comments.

We hope our answers and additions will be satisfactory responses to your questions and comments.

Thank you for considering this manuscript.

  1. The authors have addressed the majority of my comments in a satisfactory manner, and the revised manuscript shows substantial improvement in clarity, structure, and theoretical grounding. However, one important structural issue remains: the hypotheses are embedded within the Introduction. This is not in line with standard academic conventions. Hypotheses should be moved to a distinct section following the Introduction, typically titled “Research Objectives and Hypotheses” or similar. Including them within the Introduction disrupts the narrative flow and causes confusion in later references to the hypotheses. I recommend consolidating the hypotheses into a single, clearly numbered set and presenting them in a dedicated section outside the Introduction.

Reviewer 2 Report

Comments and Suggestions for Authors

Thank you for addressing the earlier feedback.

Reviewer 3 Report

Comments and Suggestions for Authors

Authors have made changes according to the previous suggestions. The article is now acceptable for publication.

Author Response

Thank you for considering this manuscript.